# Optimization of ultrasonic-assisted extraction of phenolic compounds from *Clinacanthus nutans* using ionic liquid (ILs) binary solvent: Application of Peleg's model and response surface methodology

Tong Yen Suan[1], Saiful Irwan Zubairi[1]*, Ruth Naomi Manuel[1], Zalifah Mohd Kasim[1], Nur Huda-Faujan[2]

**1** Department of Food Sciences, Faculty of Science & Technology, Universiti Kebangsaan Malaysia, UKM Bangi, Selangor Darul Ehsan, Malaysia, **2** Food Biotechnology Programme, Faculty of Science and Technology, Universiti Sains Islam Malaysia, Nilai, Negeri Sembilan, Malaysia

* saiful-z@ukm.edu.my

## Abstract

*Clinacanthus nutans* (Sabah snake grass) is widely recognized for its pharmacological properties, particularly its high phenolic content and antioxidant activity. However, the optimization of its ultrasonic-assisted extraction (UAE) remains underexplored. This study aims to enhance the extraction efficiency of phenolic compounds from *Clinacanthus nutans* leaves using ionic liquid (IL) binary solvents, with optimization based on Peleg's model and Response Surface Methodology (RSM). Peleg's model was used to determine the optimal extraction time, while RSM with a Central Composite Rotatable Design (CCRD) was applied to evaluate the effects of ultrasonic frequency (40–60 kHz) and the ratio of ILs to water (2:8, 5:5 and 8:2) on total phenolic content (TPC), DPPH radical scavenging activity, and Ferric Reducing Antioxidant Power (FRAP). The experimental results were statistically analyzed using ANOVA, model fitting, and desirability functions. Peleg's model indicated that the predicted maximum total phenolic content (TPC) of $42.556 \pm 0.0003$ mg GAE/g was achieved at an ultrasonic frequency of 50 kHz within 3 hours, making this duration as the predictive model benchmark for further optimization. The optimal extraction conditions were identified as an ultrasonic frequency of 60 kHz and an IL-to-water ratio of 2:8, yielding a maximum TPC of $0.01 \pm 7.97 \times 10^{-5}$ mg GAE/g, DPPH antioxidant activity of $95.08 \pm 0.57\%$, and FRAP antioxidant capacity of $6.31 \pm 0.10$ mg AEAC/g. Peleg's model inadequately predicted the best exhaustive extraction time prior to RSM leading to a low TPC value throughout the optimization process while maintaining high in antioxidant efficacy. However, the use of IL binary solvents significantly enhanced the release of phenolic compounds compared to conventional solvents, demonstrating their potential as a green extraction alternative. This study highlights the

**Data availability statement:** All relevant data are within the paper and its Supporting Information files.

**Funding:** We express gratitude for the funding provided by Universiti Kebangsaan Malaysia (UKM) under grant ST-2023-043, which enabled us to conduct this study. Furthermore, we extend our appreciation to the Department of Food Sciences, Faculty of Science and Technology, UKM Bangi, for granting us access to their laboratory facilities. The funder had no role in study design, data collection and analysis, decision to publish, or preparation of the manuscript.

**Competing interests:** The authors have declared that no competing interests exist.

effectiveness of ultrasonic-assisted extraction combined with IL binary solvents for maximizing the recovery of bioactive compounds from *Clinacanthus nutans* leaves. The optimized extraction method can be beneficial for pharmaceutical, nutraceutical, and functional food industries. Future research should focus on identifying specific phenolic compounds using High-Performance Liquid Chromatography (HPLC), combined kinetic and diffusion equilibrium model and further refining process optimization parameters (e.g., longer concoction duration) to enhance yield efficiency.

## 1. Introduction

The Sabah snake grass, scientifically known as *Clinacanthus nutans* is a plant used for therapeutic purposes and renowned for its remarkable therapeutic properties. It belongs to the *Acanthaceae* family and is commonly found across Southeast Asia, particularly in Malaysia, Indonesia, Brunei, the Philippines, and Thailand [1,2]. The plant is distinguished by its elongated, pointed green leaves covered with fine hairs. It thrives in lowland areas with moderate moisture and is propagated using stem cuttings. *Clinacanthus nutans* can grow to a height of 1–3 meters [3].

The leaves of *Clinacanthus nutans* have been used extensively in traditional medicine to treat a variety of conditions, including skin conditions, insect bites, and inflammation. Scientific research has confirmed that these leaves exhibit potent anti-inflammatory, antioxidant, antiviral, and antimicrobial properties [2,4]. Traditionally, the leaves were boiled and consumed as an herbal drink, believed to be effective against cancers such as colon, lung, and breast cancer. Additionally, the plant is known to address numerous medical conditions, such as excessive blood pressure, kidney difficulties, and diabetes, while also helping to reduce the risk of clogged arteries, detoxify the body, and alleviate fatigue [5].

Research conducted by the Guangdong Academy of Agricultural Sciences and the College of Horticulture at South China Agricultural University has confirmed the medicinal value of *Clinacanthus nutans*. The findings highlight its potential in preventing cancer and cardiovascular diseases, attributed to its high flavonoid content [3]. Additionally, the plant is rich in essential nutrients, including phosphorus, calcium, magnesium, iron, vitamin C, protein, and fibre, which enhance its ability to combat free radicals effectively. The acceptance of *Clinacanthus nutans* within communities has increased due to its rich antioxidant content and potential as an affordable alternative to costly medications. Its high levels of antioxidants offer significant health benefits by protecting body cells from damage caused by free radicals, which are linked to diseases such as cancer, heart conditions, and premature ageing [6]. Furthermore, in rural areas with limited financial resources, the high cost of doctor-prescribed medications makes *Clinacanthus nutans* an appealing alternative, as it is readily available in nature [3,7].

*Clinacanthus nutans* has garnered significant attention in medical research and scientific studies due to its promising therapeutic potential. Scientific evidence supporting its effectiveness in treating various diseases has strengthened trust in

its medicinal properties. As a result, *Clinacanthus nutans* is increasingly recognized and utilized as a valuable source of alternative medicine, both for addressing specific ailments and as a complementary approach to modern treatments [8]. Similarly, *Strobilanthes crispus* has been established as a benchmark due to its significant phenolic content and antioxidant activity, which have been well-documented in various studies which are comparable to or even greater than those of *Clinacanthus nutans*. One notable study demonstrated that *Strobilanthes crispus* effectively inhibits DPPH free radicals, demonstrating high antioxidant potential with an $IC_{50}$ value of 5.44 µg/mL, which supports the assertion that its antioxidant activity is comparable to or greater than that of *Clinacanthus nutans* [9]. Additionally, the phytochemical analysis of S. crispus has revealed a diverse array of phenolic compounds, notably flavonoids and phytosterols, which are acknowledged for their antioxidant properties [10,11]. In terms of extraction and subsequent biological activity, studies have highlighted that *Strobilanthes crispus* can induce apoptosis in cancer cell lines, indicating its bioactive potential linked to its high phenolic content [12,13]. The extraction methodologies, particularly ultrasonic extraction combined with enzymatic processes, have been shown to maximize the yield of bioactive compounds from *Strobilanthes crispus*, which can then be used to draw parallels to the extraction process of *Clinacanthus nutans* [14].

Peleg's model is instrumental in optimizing extraction processes for bioactive compounds by providing a quantitative framework for understanding the kinetics of extraction over time. In the context of extracting phenolic compounds from *Clinacanthus nutans* leaves using ultrasonic-assisted extraction (UAE), Peleg's equation enables the determination of key parameters, specifically the rate constant ($K_1$) and capacity constant ($K_2$). These parameters are crucial as they signify the timeframe at which no significant additional phenolic compounds can be extracted, thereby enhancing efficiency and sustainability in the extraction process while minimizing energy and solvent waste [15].

Thus, Peleg's model provides a semi-empirical framework for understanding extraction kinetics, which is particularly valuable for establishing a consistent extraction time that maximizes resource efficiency. By leveraging Peleg's model to delineate the extraction kinetics, we can identify specific points where yields become negligible and thus mitigate the risk of further extraction processes that do not yield substantial benefits. This not only streamlines the experimental process reducing the number of runs required but also grounds the later phases of the RSM in validated parameters that reflect a fixed and efficient extraction window [16,17]. In contexts where ultrasonic cavitation impacts extraction, Peleg's model proves particularly beneficial as it integrates the complexities introduced by such physical phenomena. Studies demonstrate that ultrasonic-assisted extraction enhances mass transfer by breaking down tissue structures and increasing solvent penetration, while also influencing the kinetics of extraction, making Peleg's model suitable to describe these enhanced kinetics [17–19]. Thus, the model allows for a refined approach to defining resource-efficient extraction times, which can enhance process designs when using RSM to optimize extraction parameters [20]. Furthermore, Peleg's model has been shown to deliver high correlation coefficients aligning closely with empirical data across several food extraction process, affirming its capability to serve as a reliable predictive tool for extraction kinetics [17,21]. The extraction of *Clinacanthus nutans* using ultrasonic techniques remains an area that has not been extensively explored. Although ultrasonic methods are widely employed for extracting bioactive compounds from various plants, their application to *Clinacanthus nutans* has not been thoroughly studied. Ultrasonic extraction, known for its short processing time, uses ultrasonic waves to enhance the extraction of bioactive compounds from Sabah snake grass. Variables including temperature, extraction duration, solid-to-liquid ratio, and solvent type significantly impact the yield of bioactive compounds and the bioactivity of *Clinacanthus nutans* leaves. Therefore, this study aims to determine the total exhaustive extraction time using Peleg's model and to identify the optimum extraction parameters for antioxidant activity characteristics through ultrasonic extraction, employing response surface methodology (RSM).

Despite the growing scientific interest in *Clinacanthus nutans*, limited studies have systematically optimized its extraction conditions, especially using green solvents like DES in combination with UAE. Most existing research has focused on conventional solvent systems, which may pose environmental or safety concerns. Furthermore, while UAE has been used for other botanicals, its application to *Clinacanthus nutans* remains underexplored, particularly in conjunction

with Peleg's model for extraction kinetics and RSM for process optimization. Hence, this study aims to fill this gap by integrating a kinetic modelling approach with process optimization using environmentally friendly DES to enhance the extraction of phenolic compounds and antioxidant activity from *Clinacanthus nutans* leaves.

## 2. Materials and methods

### 2.1 Chemicals

Choline chloride (ChCl) was combined with 1,4-butanediol to prepare a deep eutectic solvent (DES) for extracting bioactive compounds from *Clinacanthus nutans* leaves. The total phenolic content (TPC) was determined using the Folin-Ciocalteu reagent (Chemiz, Malaysia) and 7.5% anhydrous sodium carbonate ($Na_2CO_3$). The free radical scavenging activity was assessed using the DPPH assay, which involved 2,2-diphenyl-1-picrylhydrazyl (DPPH) and methanol. For the ferric reducing antioxidant power (FRAP) assay, 2,4,6-tri-(2-pyridyl)-s-triazine (TPTZ), ferric (III) chloride hexahydrate ($FeCl_3 \cdot 6H_2O$), and sodium acetate buffer were used. Gallic acid served as the standard for the TPC assay, while ascorbic acid was used as the standard for both the DPPH and FRAP assays. All chemicals used were of analytical grade and were procured from Sigma Aldrich, USA.

### 2.2 Materials

A total of 2.5 kg of Sabah snake grass (*Clinacanthus nutans*) leaves was obtained from TKC Herbal Nursery Sdn. Bhd., with one-month-old leaves selected for the extraction process. Additionally, 2.5 kg of *Strobilanthes crispus* (glass splinter) leaves were sourced from *Rumah Tumbuhan* UKM, Bangi, Selangor, Malaysia, to serve as a benchmark for the study. Upon the arrival of both leaf types, a visual classification was conducted, and leaves with specific characteristics were selected for extraction. The selected *Clinacanthus nutans* and *Strobilanthes crispus* leaves were light green, less than 5 cm in size, thin, soft, and smooth to the touch. Based on Fig 1, the leaf labelled with index 10 was utilized in this study.

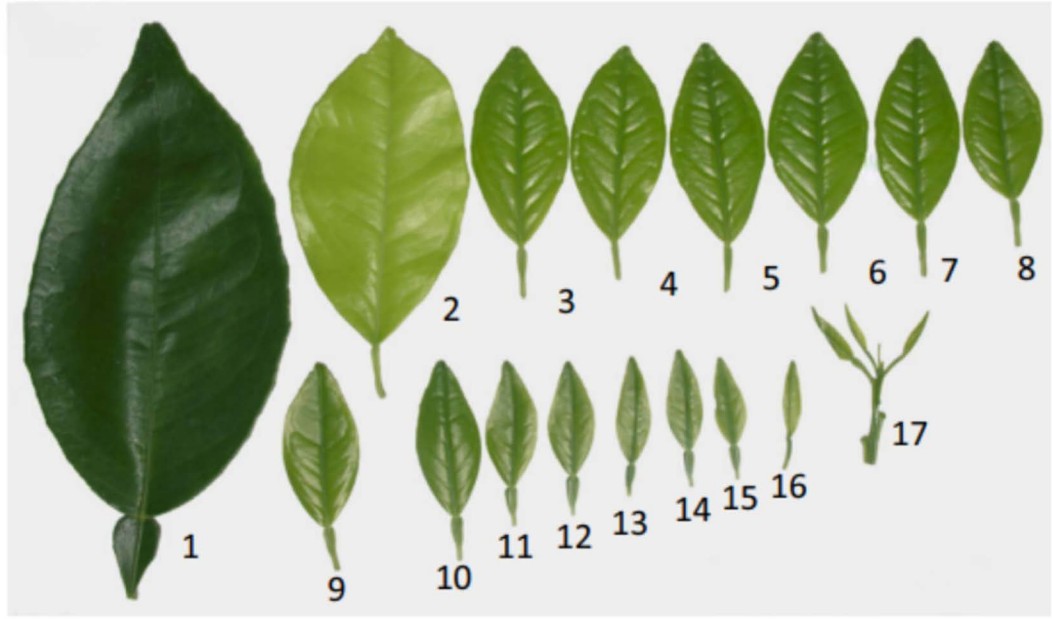

**Fig 1. Selected Sabah snake grass maturity index prior to the ultrasonicated extraction process.** #1: mature leaf; #2: fully developed leaf; #3-#16: immature leaves; #17: youngest leaf at the apical meristem.

## 2.3 Sample preparation

The *Clinacanthus nutans* leaf samples were dried in an oven (INB 500, Incubator Memmert, Germany) at 40°C for 48 hours. To minimize thermal degradation, the leaf samples were dried at a low temperature of 40°C for 48 hours, which is commonly reported in literature as a safe range for preserving heat-sensitive phytochemicals. This temperature was carefully selected to avoid enzymatic activity and microbial growth while maintaining compound integrity, particularly for phenolics and flavonoids [22–24]. After drying, the leaves were ground using a Waring Commercial Blender for 1 minute with 10-second intervals to obtain a fine powder and sieved by a sieve filter. The resulting dry and fresh fine leaf powder was kept in containers for later extraction procedures at ambient temperature (25°C). Similarly, the *Strobilanthes crispus* leaves were separated from their stems and washed thoroughly under running water until clean. Once free of impurities, the leaves were dried in an oven (INB 500, Incubator Memmert, Germany) at 40°C. The dried leaves were then ground using a Waring Commercial Blender for 1 minute with 10-second intervals to produce a fine powder and sieved by a sieve filter. The prepared dry and fresh fine leaf powder was stored in universal containers at room temperature (25°C) for further analysis.

## 2.4 Preparation of deep eutectic solvent (DES)

A molar ratio of 1:5 was used to combine choline chloride (ChCl) and 1,4-butanediol to create the deep eutectic solvent (DES), following the method described by Bi et al. (2013) [25]. The mixture was stirred with a glass rod (Marienfeld-Superior, Germany) at 80°C until a uniform mixture was created. After cooling until the room temperature was 25–27°C, the solution was stored in a Schott bottle, as outlined by Zetty et al. (2015) [26]. The prepared DES was then used in the ultrasonic extraction process by combining it with *Clinacanthus nutans* leaves.

## 2.5 Experimental design

The response surface methodology with central composite design (RSM-CCRD) was employed to optimize the extraction process for determining the total phenolic content and antioxidant capacity in *Clinacanthus nutans* leaves. Table 1 shows the coded and uncoded levels of independent variables that used in the response surface methodology. A three-level central composite design (CCRD) was used, with dependent and independent variables input into a computer system to automatically generate the number of experiments. According to Table 2, a total of 13 experiments were conducted, including 4 axial points and 5 centre points. The two independent variables in this study are ultrasonic frequency (kHz) ($X_1$) and the variation ratio in the concentration of ionic liquid (ILs) and water (v/v) ($X_2$). These variables will be related to the total phenolic content ($Y_1$), DPPH antioxidant capacity ($Y_2$), and FRAP ($Y_3$). All experiments were performed in triplicate. The optimization of the extraction process was carried out using ultrasonic methods to enhance the release and diffusion of flavonoid components (orientin and vitexin).

## 2.6 Peleg mathematical model for ultrasonic-assisted extraction (UAE) process

The Peleg model is a valuable mathematical equation for predicting the optimal concentration of a substance at a specific time [27]. This model involves two constants, $K_1$ and $K_2$, which are derived from the equation $y = mx + c$, where $K_1$ represents the y-intercept, and $K_2$ corresponds to the slope. The Peleg model can be expressed mathematically as follows:

**Table 1. Coded and uncoded levels of independent variables used in the response surface methodology.**

| Independent variables | Symbol | Level | | |
|---|---|---|---|---|
| | | Low (−1) | Middle (0) | High (+1) |
| Ultrasonic frequency (kHz) | $X_1$ | 40 | 50 | 60 |
| Variation ratio of the concentration of ionic liquid and water (v/v) | $X_2$ | 2:8 | 5:5 | 8:2 |

Table 2. Experimental design and the value for response surface methodology.

| Sample | Independent variables (Factor) | |
| --- | --- | --- |
| | Ultrasonic frequency (kHz) | Variation ratio of the concentration of ionic liquid and water (v/v) |
| | $X_1$ | $X_2$ |
| 1 | 50.00 (0) | 5:5 (0) |
| 2 | 35.86 (−1.414) | 5:5 (0) |
| 3 | 60 (1) | 2:8 (−1) |
| 4 | 60 (1) | 8:2 (1) |
| 5 | 50 (0) | 5:5 (0) |
| 6 | 50 (0) | 5:5 (0) |
| 7 | 40 (−1) | 2:8 (−1) |
| 8 | 50 (0) | 5:5 (0) |
| 9 | 50 (0) | 5:5 (0) |
| 10 | 50 (0) | 9.24:0.76 (1.414) |
| 11 | 64.14 (1.414) | 5:5 (0) |
| 12 | 50 (0) | 0.76:9.24 (−1.414) |
| 13 | 40 (−1) | 8:2 (1) |

$$C(t) = \frac{C_o + t}{K_1 + K_2 xt} \tag{1}$$

where the extract concentration (mg/g) at time t (minutes), $C_o$ is the concentration at t = 0 (mg/g), $K_1$ is the Peleg constant 1 (min·g/mL), and $K_2$ is the Peleg constant 2 (mL/g). If the concentration ($C_o$) at t = 0 is 0, the linear equation is as follows:

$$\frac{t}{C(t) - C_o} = K_1 + K_2 \times t \tag{2}$$

The Peleg constant 1 ($K_1$) is associated with the concentration ($B_o$) at a specific time (t = $t_o$). In equation (3), the relationship can be expressed as follows:

$$B_o = \frac{1}{K_1} \tag{3}$$

The constant $K_2$ is related to the maximum extract concentration. When $t \to \infty$ (the extract reaches equilibrium, where the solute concentration stabilizes within a given extract volume), Equation (4) defines the relationship between $K_2$ and the extract concentration as follows:

$$C(t) \to \infty = Ce = \frac{1}{K_1} \tag{4}$$

## 2.7 Ultrasonic-assisted extraction (UAE) approach

The ultrasonic solid-liquid extraction process was adapted from the study by Zhang et al. (2011) [28], with slight modifications. The extraction of polyphenols and antioxidants from *Clinacanthus nutans* leaves was performed using an ultrasonic device (Model JY96-IIN, Lichen, China). A total of 10 g of *Clinacanthus nutans* leaf samples was mixed with 100 mL of an

ionic solution, prepared by dissolving the deep eutectic solvent (DES) in deionized water at varying ratios of concentration of ionic liquid (IL) to water (2:8, 5:5 and 8:2). The mixture was subjected to ultrasonic treatment at a specified frequency ranging from 40 kHz to 60 kHz for a fixed duration, as determined by the Peleg mathematical model. The UAE process was conducted under ambient temperature conditions (25–27°C). The ultrasonic device used was equipped with an internal cooling system, and the extraction vessel was partially immersed in a temperature-controlled water bath to dissipate heat generated by ultrasonic cavitation. This setup ensured that the temperature remained stable throughout the 3-hour extraction period, minimizing any risk of thermal degradation. During the process, 1 mL of the extract was sampled every 20 minutes for a total duration of 3 hours and filtered using a muslin bag. The filtered extract was then analyzed for flavonoid components using response surface methodology (RSM), as described by Aybastier et al. (2013) [29]. The entire extraction process was conducted in triplicate ($n = 3$) to ensure reproducibility.

## 2.8 Total phenolic content (TPC) analysis

A total of 1 mL of extract was placed in a test tube and mixed with 5 mL of Folin-Ciocalteu reagent, which had been diluted 10-fold with distilled water. After 5 minutes, 4 mL of 7.5% (w/v) sodium carbonate ($Na_2CO_3$) was added. The mixture was shaken and incubated in the dark at room temperature (27°C) for 2 hours [30]. The absorbance was measured at 765 nm using a microplate spectrophotometer (Epoch™ Microplate Spectrophotometer, BioTek Instruments, USA) [31].. Each sample was measured in triplicate to calculate the average absorbance ± standard deviation ($n = 3$). Gallic acid was used as the standard solution for this test. Gallic acid values of 0, 20, 40, 60, 80, and 100 ppm were used to create a standard curve [30]. Gallic acid equivalents (GAE) were used to express the sample's total phenolic content (TPC) using the following formula:

$$\text{Total phenolic content} = \frac{R \text{ x } DF \text{ x } TV}{SV \text{ x } wt \text{ x } 10,000}$$

(5)

where R is the concentration from the standard curve (µg/mL), DF is the dilution factor (100), TV is the total sample volume (µL), SV is the sample volume used (µL), and wt is the extract mass (g).

## 2.9 DPPH free radical scavenging activity

Following the method described by Alara et al., (2017) [32], a 0.15 mM DPPH (2,2-diphenyl-1-picrylhydrazyl) solution was prepared as the stock solution and freshly made before each use. In a volumetric flask, 5.9 mg of DPPH powder was dissolved in 100 mL of methanol to create the DPPH reagent and mixed thoroughly. For this study, 1 mL of each sample solution was pipetted into Eppendorf tubes, followed by the addition of 2 mL of the 0.15 mM DPPH solution. The mixture was shaken and left to incubate in the dark at room temperature (27°C) for 30 minutes. The absorbance of each sample was then measured at 517 nm using a microplate spectrophotometer (Epoch™ Microplate Spectrophotometer, BioTek Instruments, USA). The DPPH free radicals reacted with antioxidant compounds in the sample, resulting in a colour change from dark purple to light yellow. All extract samples were analyzed in triplicate, and absorbance readings were averaged. Ascorbic acid was used as the blank. The percentage of DPPH free radical scavenging activity was calculated using the following formula:

$$\text{DPPH scavenging activity } (\%) = \frac{(A_1 - A_2)}{A_1} x \text{ } 100\%$$

(6)

where $A_1$ represents the absorbance of the control, and $A_2$ represents the absorbance of the sample.

## 2.10 Ferric reducing antioxidant power (FRAP)

A 20 mM ferric chloride hexahydrate ($FeCl_3 \cdot 6H_2O$) solution, a 10 mM 2,4,6-tri-(2-pyridyl)-s-triazine (TPTZ) solution in 40 mM HCl, and a 30 mM acetate buffer ($CH_3COONa$) at pH 3.6 were prepared separately. The acetate buffer was made

by dissolving 3.1 g of sodium acetate trihydrate in 16 mL of glacial acetic acid and mixing it with distilled water in a 1:1 ratio. The FRAP reagent was then prepared by mixing 50 mL of 30 mM acetate buffer, 5 mL of 10 mM TPTZ solution, and 5 mL of 20 mM $FeCl_3 \cdot 6H_2O$ solution in a volumetric ratio of 10:1:1. For the analysis, 0.1 mL of extract was combined with 1 mL of freshly prepared FRAP reagent. The mixture was incubated at room temperature in the dark for 30 minutes, after which its absorbance was measured at a wavelength of 593 nm using a microplate spectrophotometer (Epoch™ Microplate Spectrophotometer, BioTek Instruments, USA). A standard curve was generated using solutions with concentrations of 0, 20, 40, 60, 80 and 100 ppm. Each absorbance reading was measured in triplicate ($n = 3$) to calculate the average ± standard deviation. Ascorbic acid equivalent antioxidant capacity (AEAC), measured in mg of AEAC/g extract, was used to express the ferric reducing antioxidant power (FRAP) value, calculated using the following formula:

$$\text{Ferric reducing antioxidant power (mg extract AEAC/g)} = \frac{R \text{ x } DF \text{ x } TV}{SV \text{ x } wt \text{ x } 10,000} \tag{7}$$

where R is the concentration from the standard curve (µg/mL), DF is the dilution factor (100), TV is the total sample volume (µL), SV is the sample volume used (µL), and wt is the extract mass (g).

## 2.11 Statistical analysis

Every analysis was carried out in triplicate, and the mean ± standard deviation is how the results are displayed. The data were analyzed using Minitab software version 19.0. Two-way ANOVA tests were conducted at a 95% significance level ($p < 0.05$) to identify significant differences between samples. The total phenolic content and antioxidant activity of *Clinacanthus nutans* leaves were recorded as the average reading ± standard deviation. Design Expert software (Version 6.0.10, Stat-Ease Inc., Minneapolis) was used for the central composite design (CCRD) and to analyze the experimental data within the framework of response surface methodology (RSM).

## 3. Results and discussion

### 3.1 Peleg's model responses

The Peleg's mathematical model was used theoretically to determine the maximum extraction yield and the required exhaustive time of extraction. The kinetic profile for the phenolic compound extraction process from *Clinacanthus nutans* leaves using ultrasonic extraction was fitted to this model [33]. The regression line shown in Fig 2 was plotted to determine the maximum extraction yield ($K_1$) for the ultrasonic extraction of *Clinacanthus nutans* leaves. The significance of the Peleg constant value $K_1$ (the y-intercept) is closely linked to the extraction yield. A lower $K_1$ value indicates a higher extraction yield [34]. Additionally, the logarithmic equation in Fig 3 was used to determine the extraction exhaustive time.

   Referring to Fig 3, the extraction curve profile of the total phenolic content for *Clinacanthus nutans* leaves demonstrates the extraction rates at three ultrasonic frequencies: 40 kHz, 50 kHz and 60 kHz. The extraction rate is influenced by the movement and mass transfer within the solvent [35,36]. Table 3 presents the predicted maximum values for the ultrasonic extraction process based on the Peleg mathematical model at the three different ultrasonic frequencies. A predicted 3-hour extraction exhaustive time was selected for the response surface methodology (RSM) optimization process. This duration was chosen because, at 50 kHz, the predicted total phenolic content achieved was the highest (42.556 ± 0.0003 mg GAE/g) compared to the 40 kHz and 60 kHz frequencies. According to Melecchi et al. (2006) [36], a longer extraction time can improve the extraction yield. Therefore, a 3-hour extraction exhaustive time was selected to optimize the antioxidant activity and overall phenolic content.

### 3.2 3-D response surface model fitting

The ultrasonic extraction process of herbal leaves plays a crucial role in enhancing efficiency while reducing extraction time to maximize yield [36]. To optimize this process for extracting total phenolic content and antioxidant activity (DPPH

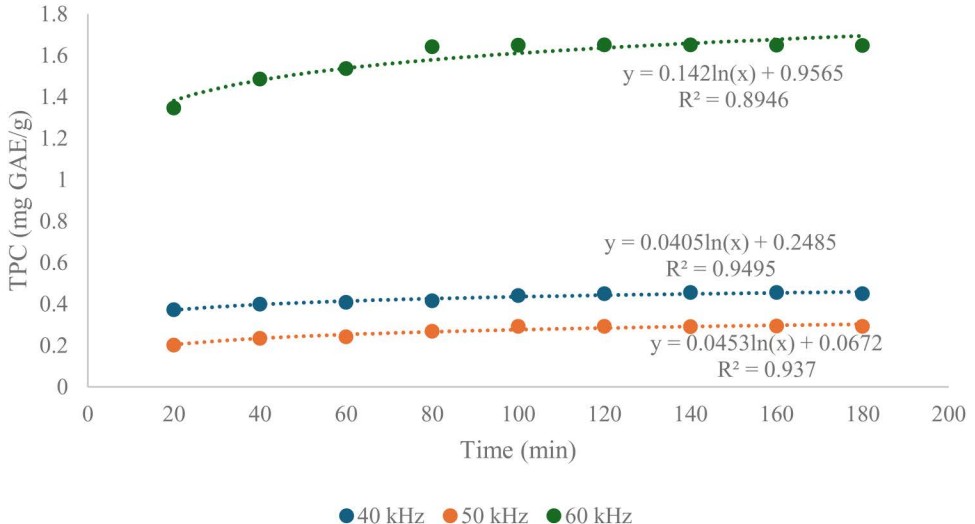

**Fig 2. Regression lines of Peleg's mathematical model fitted to the TPC of Sabah snake grass extract at different ultrasonic frequencies (40, 50, and 60 kHz) over time (20–180 min).** TPC is expressed as mg GAE per gram of dry sample. The experimental data were modelled using a logarithmic form of Peleg's equation, showing the relationship between extraction time and phenolic yield. Among the three frequencies, 60 kHz produced the highest phenolic extraction efficiency, achieving a maximum TPC of ~1.7 mg GAE/g. The corresponding model ($y = 0.142\ln(x) + 0.9565$; $R^2 = 0.8946$) indicates a strong correlation between extended sonication time and higher TPC. In contrast, 40 kHz ($y = 0.0405\ln(x) + 0.2485$; $R^2 = 0.9495$) and 50 kHz ($y = 0.0453\ln(x) + 0.0672$; $R^2 = 0.937$) treatments yielded significantly lower TPC values, reflecting reduced mass transfer and cell disruption at lower frequencies.

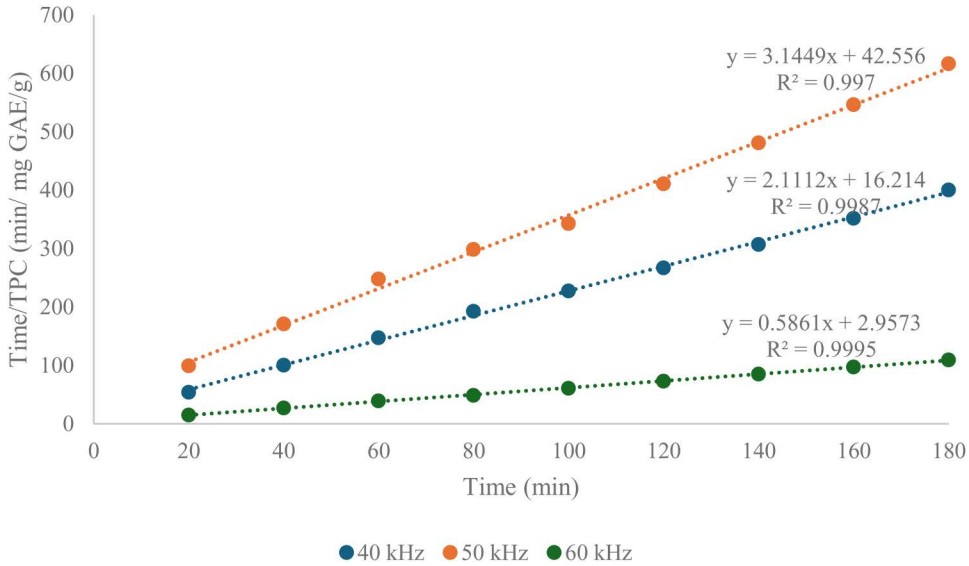

**Fig 3. Linearized Peleg's model fitting of TPC extraction kinetics for Sabah snake grass at different ultrasonic frequencies (40, 50, and 60 kHz).** The parameter Time/TPC (min/mg GAE/g) is plotted against time (min) to evaluate the extraction kinetics based on Peleg's second-order model. Linear regression equations and their coefficients of determination ($R^2$) are provided for each frequency. The results reveal excellent model fits for all frequencies, with $R^2$ values exceeding 0.99, indicating the suitability of Peleg's model in describing the extraction behavior. Notably, the slope values related to the Peleg rate constant ($k_1$) differ substantially among frequencies, with 60 kHz ($y = 0.5861x + 2.9573$; $R^2 = 0.9995$) showing the lowest slope, signifying the highest extraction rate and most efficient phenolic recovery. In contrast, 50 kHz ($y = 3.1449x + 42.556$; $R^2 = 0.997$) demonstrated the slowest extraction kinetics, as reflected by the steepest slope and highest time requirement per unit of TPC extracted.

**Table 3. Dependent variables in Peleg's mathematical model.**

| Dependent variables | Frequency (kHz) | | |
|---|---|---|---|
| | 40 | 50 | 60 |
| Extraction rate (mg GAE/g·min) | $2.1112 \pm 0.005^b$ | $3.1449 \pm 0.003^a$ | $0.5861 \pm 0.019^c$ |
| Predicted maximum extraction yield (TPC: mg GAE/g) | $16.214 \pm 0.0003^a$ | $42.556 \pm 0.0003^a$ | $2.957 \pm 0.0011^b$ |
| Predicted time of exhaustive extraction (min) | $77.58 \pm 23.08^c$ | $166.47 \pm 17.56^b$ | $302.07 \pm 10.15^a$ |

Values [a-c]: Different letters within the same row indicate significant differences ($p < 0.05$).

and FRAP) from *Clinacanthus nutans* leaf extract, a three-level composite central design (CCRD) with two independent variables was employed. According to the Peleg mathematical model, the extraction time for all experiments in this study was set at 3 hours. The upper and lower values for all factors were +α (+α = 1.414) and -α (-α = −1.414), respectively, ensuring that all factor levels fell between these limits.

The experimental values for Total Phenolic Content (TPC), DPPH Free Radical Scavenging Activity, and FRAP (Ferric Reducing Antioxidant Power) of *Clinacanthus nutans* leaf extract are presented in Table 4. The results show that TPC, DPPH, and FRAP values for the extract ranged between (0.0035-0.0102 mg GAE/g), (43.04-95.08%), and (2.25-6.31 mg AEAC/g), respectively. These results are reported as the mean ± standard deviation from three replications. The acquired data was fitted to a polynomial equation of second order, and the regression equations in Table 5 present the models used to fit the TPC, DPPH, and FRAP values for the *Clinacanthus nutans* leaf extract, based on ultrasonic frequency ($X_1$) and the variation ratio in the concentration of ionic liquid (ILs) and water ($X_2$). According to Affam (2020) [37] and Ooi et al. (2018) [38], statistical significance is established when the p-value is less than 0.05 ($p < 0.05$). In this study, statistical results indicate that the quadratic model is suitable for TPC data ($p < 0.05$), but not for DPPH and FRAP ($p > 0.05$).

An ANOVA test was performed to assess the quality and reliability of the response surface methodology (RSM) diagnostic tool [39]. The ANOVA for the regression model shows that the model is not significant for all dependent variables ($p > 0.05$). Model fit is assessed using the lack-of-fit test and the coefficient of determination ($R^2$). The mathematical model was selected based on statistical significance at a confidence level of $p < 0.05$. The lack-of-fit values indicated that the quadratic model was statistically significant ($p < 0.05$) for TPC, but not for DPPH and FRAP ($p > 0.05$), with $R^2$ values of 0.8149, 0.0480, and 0.6120, for TPC, DPPH, and FRAP respectively. The coefficient of determination ($R^2$) represents the ratio of explained variation to total variation, serving as a measure of model goodness-of-fit [40]. A small $R^2$ value indicates a weak relationship between the dependent variables in the model, and a value approaching 1 would suggest a good fit. However, the $R^2$ values for all three dependent variables (TPC, DPPH and FRAP) were far from 1, suggesting that the model does not fit the data well for assessing the relationships between the dependent and independent variables.

Phenolic compounds, characterized by one or more hydroxyl groups attached to an aromatic hydrocarbon chain, display diverse biological activities primarily due to their antioxidant properties. Research indicates that TPC is often positively correlated with antioxidant activities across different plant extracts. For instance, studies have demonstrated that higher concentrations of phenolic compounds correspond to increased radical scavenging activity as exhibited in DPPH assays, suggesting that phenolic compounds play substantial roles in providing antioxidant effects due to their ability to donate electrons [41]. Notably, it has been shown that specific phenolic acids exhibit inhibitory effects on tyrosinase (a key enzyme in melanin synthesis) by forming hydrogen bonds with the enzyme, further emphasizing the functional relevance of phenolic structures [42]. certain phenolic compounds are linked to broader pharmacological activities such as anti-inflammatory and antimicrobial properties [43].

**Table 4. Experimental design and values of the observed response surface optimization.**

| Sample | Independent variables (Factor) | | Dependent variables (Response) | | |
|---|---|---|---|---|---|
| | Ultrasonic frequency (kHz) | Variation ratio of concentration of ionic liquid and water (v/v) | TPC (mg GAE/g) | Ultrasonic frequency (kHz) | |
| | | | | DPPH (%) | FRAP (mg AEAC/g) |
| | $X_1$ | $X_2$ | $Y_1$ | $Y_2$ | $Y_3$ |
| 1 | 50.00 (0) | 5:5 (0) | $0.0042 \pm 0.0025$ | $82.890 \pm 17.749$ | $4.2676 \pm 0.2598$ |
| 2 | 35.86 (−1.414) | 5:5 (0) | $0.0051 \pm 0.0022$ | $85.294 \pm 7.362$ | $3.8725 \pm 0.1415$ |
| 3 | 60 (1) | 2:8 (−1) | $0.0102 \pm (7.97 \times 10^{-5})$ | $95.075 \pm 0.570$ | $6.3076 \pm 0.0961$ |
| 4 | 60 (1) | 8:2 (1) | $0.0100 \pm 0.0001$ | $74.907 \pm 8.063$ | $2.5447 \pm 0.1967$ |
| 5 | 50 (0) | 5:5 (0) | $0.0047 \pm 0.0002$ | $91.597 \pm 6.013$ | $4.1448 \pm 0.2922$ |
| 6 | 50 (0) | 5:5 (0) | $0.0053 \pm 0.0002$ | $91.200 \pm 8.697$ | $4.1841 \pm 0.1589$ |
| 7 | 40 (−1) | 2:8 (−1) | $0.0073 \pm (6 \times 10^{-5})$ | $59.010 \pm 11.408$ | $3.7376 \pm 0.2416$ |
| 8 | 50 (0) | 5:5 (0) | $0.0041 \pm (3.68 \times 10^{-5})$ | $46.849 \pm 17.350$ | $2.9541 \pm 0.5273$ |
| 9 | 50 (0) | 5:5 (0) | $0.0049 \pm (8.51 \times 10^{-5})$ | $46.172 \pm 5.608$ | $3.0929 \pm 0.0720$ |
| 10 | 50 (0) | 9.24:0.76 (1.414) | $0.0040 \pm 0.0024$ | $89.169 \pm 5.141$ | $2.5339 \pm 0.1569$ |
| 11 | 64.14 (1.414) | 5:5 (0) | $0.0096 \pm 0.0004$ | $62.955 \pm 14.945$ | $3.6501 \pm 0.3656$ |
| 12 | 50 (0) | 0.76:9.24 (−1.414) | $0.0037 \pm 0.0004$ | $51.961 \pm 2.952$ | $3.4287 \pm 0.1073$ |
| 13 | 40 (−1) | 8:2 (1) | $0.0035 \pm 0.0003$ | $43.044 \pm 3.623$ | $2.2481 \pm 0.0835$ |

**Table 5. Model fitting results for TPC, DPPH and FRAP values of Sabah snake grass leaf extract with ultrasonic frequency and variation ratio of concentration of ionic liquid and water.**

| Dependent variables | Actual model | Model | Lack-of-fit | $R^2$ |
|---|---|---|---|---|
| TPC | $0.0507 - 0.001845\, X_1 - 0.0181\, X_2 + 0.000019\, X_1^2 + 0.00158\, X_2^2 + 0.000300\, X_1 X_2$ | Quadratic | $p < 0.05$ | 0.8149 |
| DPPH | $35 + 0.52\, X_1 + 43\, X_2 + 0.0011\, X_1^2 - 19\, X_2^2 - 0.35\, X_1 X_2$ | Quadratic | $p > 0.05$ | 0.0480 |
| FRAP | $0.47 + 0.026\, X_1 + 9.97\, X_2 + 0.00101\, X_1^2 - 3.21\, X_2^2 - 0.189\, X_1 X_2$ | Quadratic | $p > 0.05$ | 0.6120 |

### 3.3 The effect of the independent variables (Factors) on total phenolic content (TPC)

Fig 4 illustrates the interaction effect of the variables on the Total Phenolic Content (TPC), while Fig 5 shows the contour plot of the TPC values for *Clinacanthus nutans* leaf extract, based on variations in ultrasonic frequencies and the ratio of concentration of ionic liquid (ILs) and water. The impact of ultrasonic frequency on TPC is depicted in Fig 4. The TPC value is lowest at an ultrasonic frequency of approximately 50 kHz. As the frequency moves towards 40 kHz and 60 kHz, the TPC value increases, causing the surface plot to rise. This suggests that ultrasonic frequencies at 40 kHz and 60 kHz result in higher phenolic content compared to around 50 kHz. Additionally, as seen in Fig 5, when the ultrasonic frequency exceeds 60 kHz, the TPC value reaches its peak. The lowest TPC values are observed between 35.86 kHz and 49.00 kHz. Therefore, the optimal ultrasonic frequency for achieving maximum extraction yield in this study is 60 kHz. At this higher

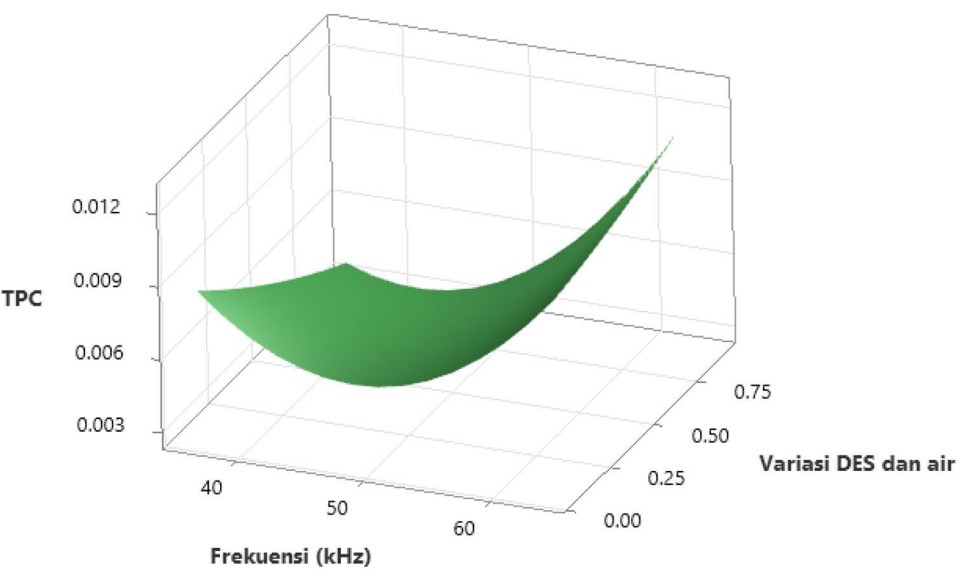

**Fig 4. 3-D surface plot showing the interactive effect of ultrasonic frequency (kHz) and the variation ratio of DES and water on the TPC of Sabah snake grass extract.** The response surface illustrates the non-linear relationship between the two independent variables and TPC yield. The plot indicates that both increasing ultrasonic frequency and optimizing the DES-to-water ratio positively influence phenolic compound recovery. The TPC initially decreases slightly with increasing frequency up to approximately 50 kHz, then increases sharply as frequency continues to rise toward 60 kHz, especially when lower proportions of water are used. Similarly, TPC increases more significantly at lower water content, highlighting the critical role of solvent composition in facilitating solute-solvent interactions and enhancing extraction kinetics. The highest TPC values are observed at the combination of higher ultrasonic frequency (around 60 kHz) and minimal water dilution.

frequency, mechanical waves generated during ultrasonic extraction promote acoustic cavitation, which breaks down the cell walls of plants when the cavitation bubbles burst on the solid matrix surface [44]. This process facilitates the phytochemicals' extraction from the vacuoles and releases those attached to the components of the cell wall [45], improving mass transfer and strengthening the solvent-plant material interaction surface area [46]. As a result, more phenolic compounds are released from the solvent matrix [15]. Kumar et al. (2021) note that ultrasonic frequencies used in bioactive component extraction typically range from 20 kHz to 120 kHz [47].

The effect of the variation ratio of the concentration of ionic liquid (ILs) and water on TPC is also evident in Figs 4 and 5. Both figures clearly show that TPC values are highest at the lowest variation ratio of ionic liquid and water (<2:8). As the ratio changes from 2:8–8:2, the TPC value decreases. Increasing the water content reduces the viscosity of a high-viscosity ionic liquid binary solvent (DES), but this also alters the physicochemical properties of the DES [48]. According to Taweekayujan et al. (2023), water molecules integrate into the DES structure, reducing viscosity and increasing TPC values [49]. However, in this study, TPC values remained insignificant ($p < 0.05$) when water content in DES increased from 20% to 80%. Water was added to the DES to enhance its polarity and lower its viscosity [50]. When water is introduced, hydrogen bonds between DES compounds are broken, and new bonds form between the DES and water [51]. As the water content in DES increases, its polarity enhances the extraction capacity for phenolic compounds [45]. The reduced viscosity of DES also leads to improved mass transfer and better diffusion of compounds [52]. Water molecules adsorb onto the DES molecular matrix, forming hydrogen bonds between ions and hydrogen bond donors [51].

### 3.4 The effect of the independent variables (Factors) on DPPH

Fig 6 illustrates the interaction effect plot of the variables on DPPH, while Fig 7 presents the contour plot of DPPH values for *Clinacanthus nutans* leaf extract, based on variations in ultrasonic frequencies and the ratio of concentration of

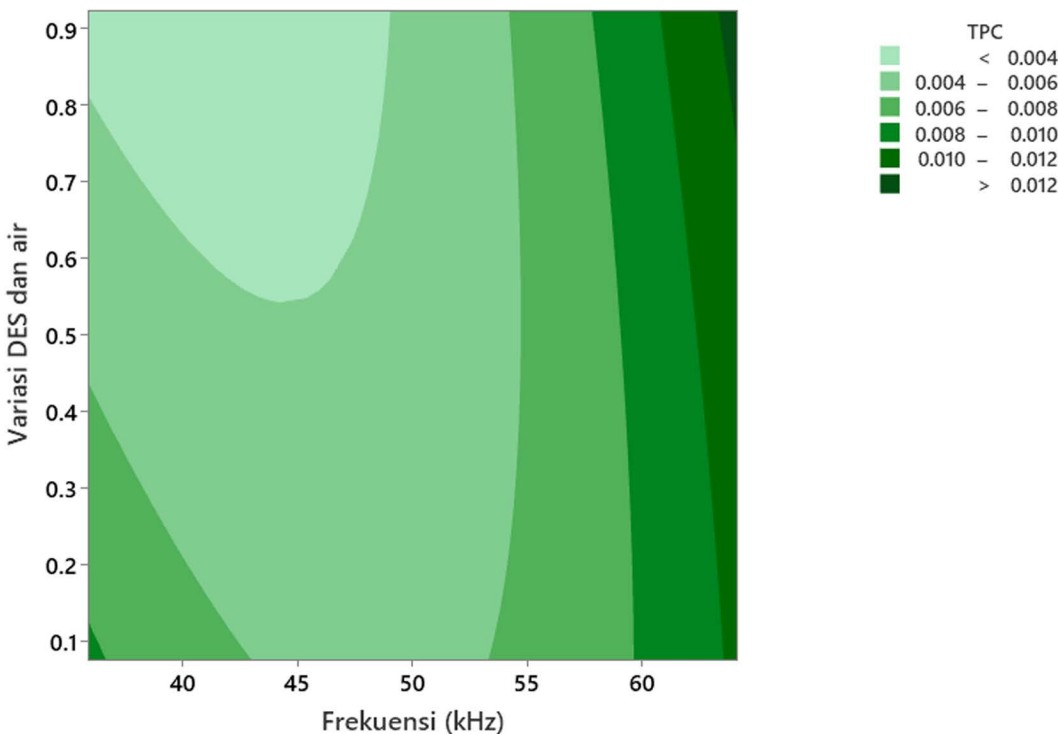

**Fig 5. Contour plot of the effect of ultrasonic frequency and the variation ratio of DES and water on TPC.** As the ultrasonic frequency increases, a moderate rise in TPC is observed, with peak regions noticeable around 55–60 kHz. Additionally, the DES-water ratio also shows a positive impact on TPC up to a certain threshold, particularly around a ratio of 0.7–0.8, after which the effect plateaus or slightly diminishes. Darker green regions in the contour represent higher TPC concentrations, exceeding 0.012, suggesting optimal combinations of higher frequencies with elevated DES content. Conversely, lower frequencies (<45 kHz) and lower DES ratios (<0.3) correspond with reduced TPC values (<0.004), as indicated by the lightest shaded areas. The gradient shifts in the plot suggest a synergistic relationship, where tuning both parameters contributes to maximizing phenolic extraction from the plant matrix.

ionic liquid (ILs) and water. From Fig 6, it is evident that the DPPH value increases as the ultrasonic frequency rises from 40 kHz to 60 kHz. Fig 7 further shows that the maximum DPPH value (greater than 76%) is achieved when the ultrasonic frequency exceeds 58.93 kHz. Phenolic compounds contribute to DPPH free radical scavenging activity due to their ability to donate hydrogen atoms and form stable intermediate radicals [53]. These compounds, characterized by one or more hydroxyl-grouped aromatic rings, exhibit reduction activity based on the number of free hydroxyl groups, which enhances their proton donation and stabilization of DPPH radicals [54].

The effect of ultrasound on the extraction process is attributed to the vibrations between the IL (DES) binary solvent and the solid matrix (Sabah snake grass) caused by ultrasonic waves [55]. The higher the ultrasonic frequency, the stronger the vibrations, which likely enhance the phenolic compound extraction process due to an increased number of cavitation bubbles [56]. At lower frequencies, cavitation bubbles may lack the energy needed to break plant cell walls, leading to a lower efficiency of antioxidant release from *Clinacanthus nutans* leaves [57]. As a result, the antioxidant content and DPPH activity are reduced. In contrast, at the optimal frequency, cavitation bubbles effectively disrupt cell walls, maximizing antioxidant activity. However, when the ultrasonic frequency is too high, cavitation bubbles form rapidly and abundantly, weakening sound energy within the liquid [58]. This can lead to excessive heat generation, potentially causing thermal oxidation or deterioration of the antioxidant compounds [59], thereby reducing DPPH free radical scavenging activity.

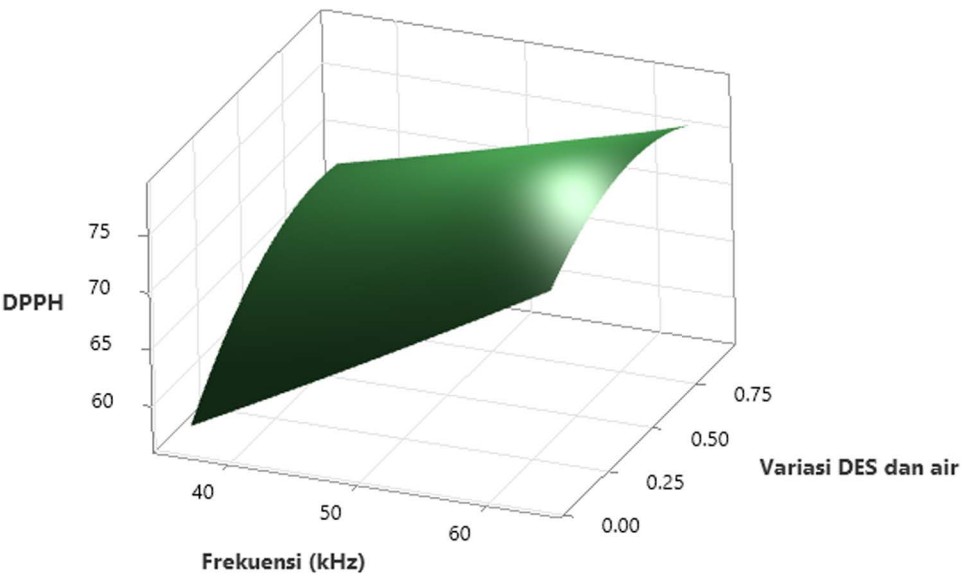

**Fig 6. 3-D surface plot showing the effect of ultrasonic frequency and the variation ratio of DES and water on DPPH radical scavenging activity.** The DPPH activity rises significantly from approximately 58% at lower frequencies (~40 kHz) and low DES content, reaching values above 75% at frequencies beyond 60 kHz and DES–water ratios exceeding 0.6. The curvature of the surface indicates a synergistic interaction between frequency and solvent composition, where neither parameter alone yields maximum DPPH values. Instead, optimal antioxidant extraction is achieved under specific combined conditions.

The effect of the variation ratio of concentration of ionic liquid (ILs) and water is also depicted in Figs 6 and 7. The antioxidant capacity of the *Clinacanthus nutans* leaf extract is highly influenced by the solvent's polarity. In Fig 6 (3-D surface plot), the maximum DPPH value is observed when the ratio of ionic liquid binary solvent to water exceeds 8:2. As shown in Fig 7 (contour plot), the DPPH value increases as the ratio of ionic liquid (ILs) and water increases from 2:8–8:2. A higher ratio of ionic liquid to water creates a significant concentration gradient between the plant cells and the surrounding solvent, promoting faster diffusion [60]. This allows more phenolic compounds to dissolve into the mixture, indirectly increasing the phenolic compound extraction yield [55]. The polarity and chemical structure of the solvent plays a crucial role in the extraction process [61]. In this study, an ionic liquid binary solvent (DES) was used. The effectiveness of the extraction process depends on the DES concentration, which directly affects DPPH free radical scavenging activity [62,63]. When the ionic liquid concentration is low, the extraction process may not be efficient enough to dissolve phenolic compounds, resulting in low antioxidant content and reduced DPPH scavenging activity [47]. On the other hand, if the ionic liquid concentration is too high, the mixture's viscosity increases, hindering solvent movement and reducing interactions between the solvent and phenolic compounds [64]. This lowers the effectiveness of the extraction process. Additionally, excessive heat generation from a high ionic liquid-to-water ratio can damage and oxidize the antioxidant compounds [65], leading to degradation due to elevated temperatures.

### 3.5 The effect of the independent variables (Factors) on FRAP

Fig 8 displays the surface interaction effect plot of the variables on FRAP, while Fig 9 shows the contour plot of the FRAP values for *Clinacanthus nutans* leaf extract, which were produced by varying ultrasonic frequencies and the ratio of concentration of ionic liquid (ILs) and water. The effect of ultrasonic frequency on FRAP is illustrated in Fig 8. As the frequency increases from 40 kHz to 60 kHz, FRAP values also increase, indicating that higher frequencies enhance antioxidant extraction efficiency in this study. This is likely due to acoustic cavitation leads to permeabilization of cell membranes,

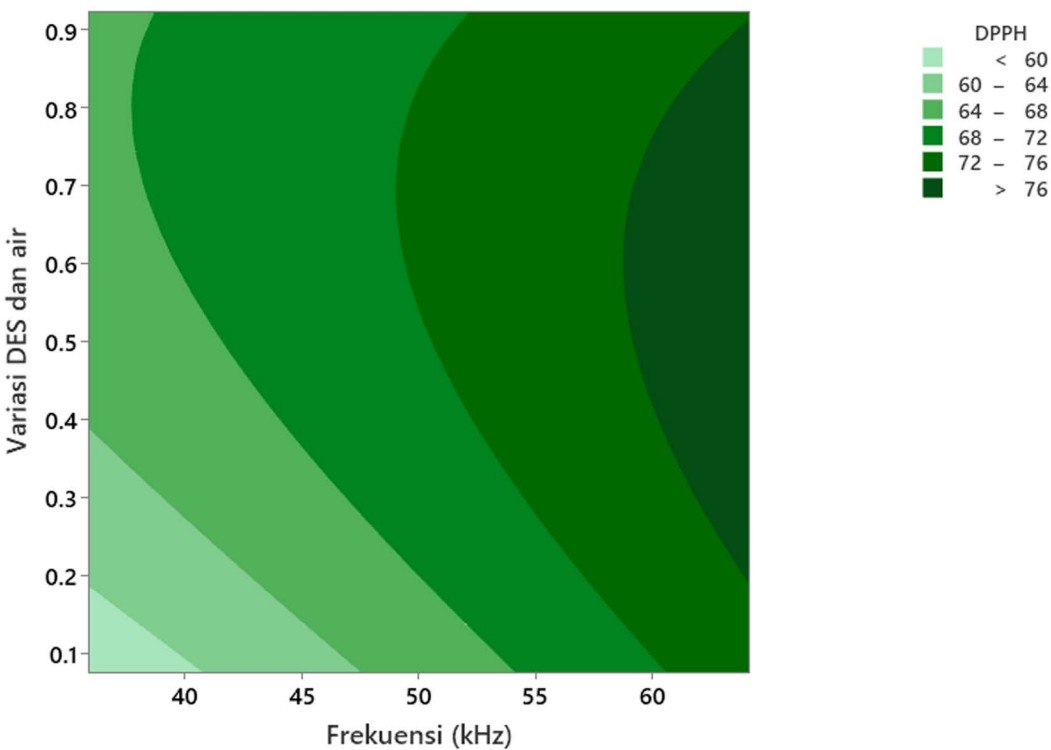

## Contour Plot of DPPH vs Variasi DES dan air, Frekuensi (kHz)

**Fig 7. Contour plot of the effect of ultrasonic frequency and variation ratio of the concentration of ionic liquid and water on DPPH.** The x-axis represents the ultrasonic frequency ranging from 37 to 61 kHz, while the y-axis indicates the variation ratio of DES and water between 0.1 and 0.9. The color gradient corresponds to DPPH values, categorized into six intervals from less than 60 to greater than 76. The contour plot shows that higher ultrasonic frequencies and higher DES concentrations lead to increased DPPH activity.

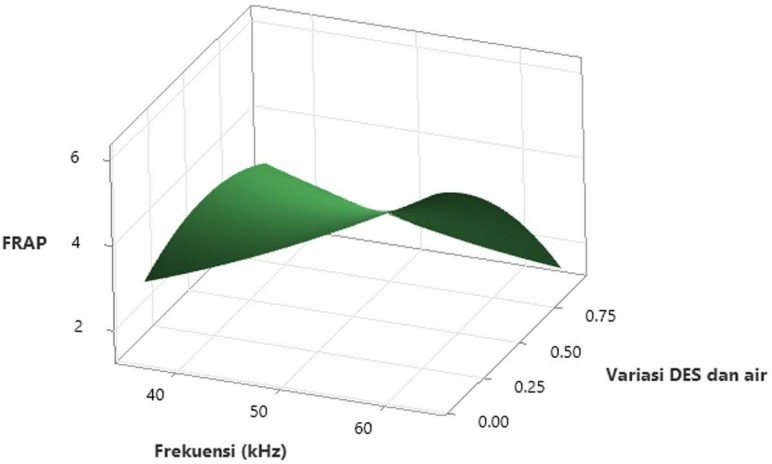

**Fig 8. 3-D surface plot of the effect of ultrasonic frequency and variation ratio of the concentration of ionic liquid and water on FRAP.** The x-axis shows ultrasonic frequency (37–61 kHz), the y-axis represents the variation ratio of DES and water (0.00–0.75), and the z-axis indicates the FRAP values. The surface plot demonstrates that both an increase in ultrasonic frequency and an optimized DES/water ratio contribute to enhancing FRAP values. A peak in antioxidant activity is observed at intermediate frequencies and DES concentrations.

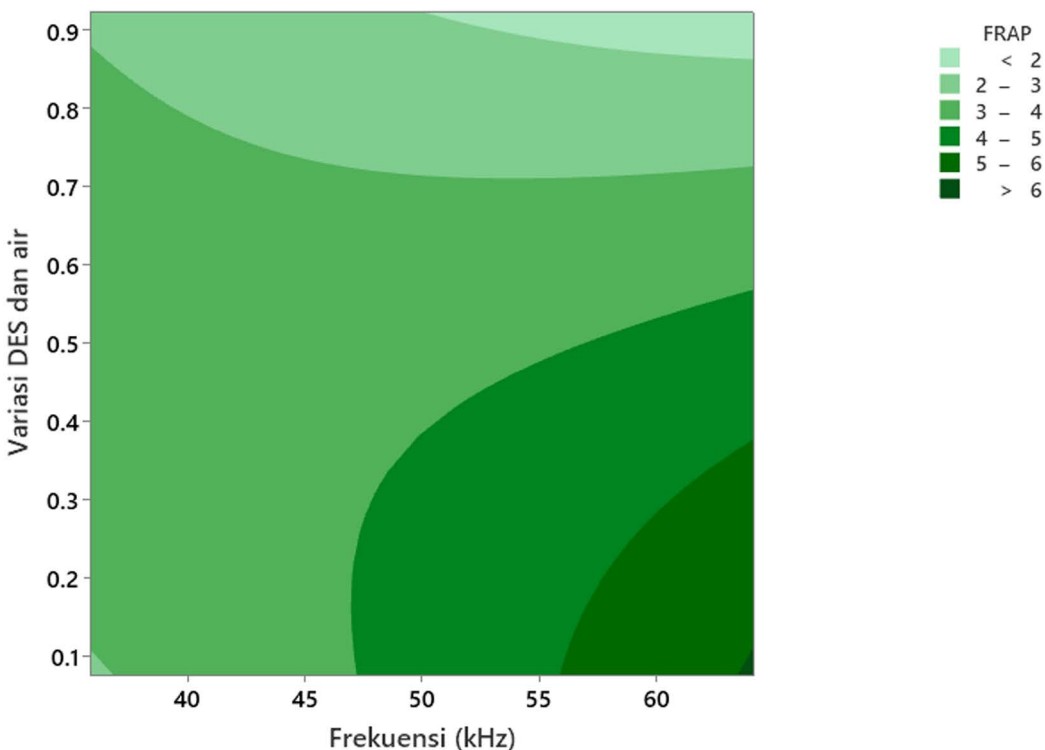

**Fig 9. Contour plot showing the effect of ultrasonic frequency (kHz) and the ionic liquid (IL)-to-water ratio on FRAP values in *Clinacanthus nutans* leaf extracts.** The contour lines represent different FRAP response levels, with darker green areas indicating higher antioxidant activity. The plot reveals that higher FRAP values (>6 mg AEAC/g) are associated with increased ultrasonic frequencies (≥60 kHz) and lower IL-to-water ratios (around 0.2), suggesting that both parameters significantly influence the efficiency of antioxidant compound extraction.

while lower frequencies might indeed form larger cavitation bubbles that could exert strong mechanical forces, these events are typically less uniformly distributed. In contrast, at 60 kHz the formation of numerous small cavitation events results in more controlled and homogeneous disruption of the cellular matrix, facilitating a more effective extraction of antioxidant compounds [66]. In our setup, the increased number of smaller, high-energy cavitation events at 60 kHz likely provided a more uniform and efficient extraction environment, resulting in higher ferric reducing antioxidant power. This supports the observed trend of enhanced FRAP values at higher ultrasonic frequencies in our results.

The effect of the variation ratio of the concentration of ionic liquid (ILs) and water on FRAP can also be observed in Figs 8 and 9. Both figures show that the highest FRAP value occurs at the lowest ratio of ionic liquid (ILs) to water (<2:8). The FRAP value decreases as the ratio changes from 2:8–8:2. Ionic liquid binary solvents (DES) with high viscosity can be made less viscous by increasing the amount of water. However, this adjustment affects the physicochemical properties of the DES [48]. The concentration of the ionic liquid binary solvent (DES) influences phenolic compounds' solubility and enhances their extraction. Increasing the concentration of DES to its maximum can cause it to become more volatile, thereby preventing the phenolic compounds from being extracted at the maximum rate [67]. Regardless the concentration, plants containing phenolic content indeed have ability to reduce ferric ion to ferrous ions [68,69].

The enhanced extraction observed under specific IL: water ratios can be fundamentally explained by considering the molecular-level solvent solute interaction mechanisms, which extend beyond mere statistical optimization. In hydrophilic ionic liquid systems, such as those based on choline chloride mixtures, the formation of extensive hydrogen bonding

networks is a well-documented phenomenon. These networks promote strong interactions with polar phenolic compounds and contribute to the disruption of plant cell wall matrices, thereby facilitating enhanced solubilization and extraction efficiency [70,71]. More specifically, the choline chloride–1,4-butanediol system exemplifies a DES where the synergistic role of hydrogen bonding is important. The bonding interactions in this system create a structured network that can efficiently interact with phenolic compounds, easing their release from plant matrices. Similar phenomena have been reported in DES-based extractions where hydrogen bonding capabilities are exploited to improve the extraction of bioactive phenolics [72,73]. In addition to the formation of hydrogen bonding networks, the system's polarity and molecular organization are key factors supporting solute dissolution, consistent with findings in other choline-based eutectic mixtures studied via experimental and computational approaches [63]. Furthermore, the incremental addition of water to the ionic liquid leads to a reduction in viscosity, a factor critical for promoting mass transfer and diffusion during extraction but also improves the wetting of plant tissues, thereby accelerating the release of phenolic compounds [74,75]. However, as the water content exceeds an optimal threshold, the dilution effect becomes counterproductive where excessive water disrupts the hydrogen bonding network and leads to reduced solvent polarity. This delicate balance between maintaining sufficient hydrogen bonding capacity and lowering viscosity explains why an IL: water ratio of approximately 2:8 appears to be optimal for phenolic extraction. This optimal balance ensures favourable conditions for enhancing both the TPC and antioxidant activity found in the extracts [72,73].

### 3.6 Optimum ultrasonicated extraction operational parameters

Optimization is the process of finding a balance between variables to simultaneously maximize the response values [76]. The optimal extraction conditions are determined and used to calculate the predicted values for the dependent variables (TPC, DPPH, and FRAP) using the prediction equations derived from response surface methodology. The determination of the optimum extraction parameters for *Clinacanthus nutans* leaves is based on achieving the highest desirability values. Fig 10 and Table 6 presents the optimal conditions for TPC, DPPH, and FRAP, along with their corresponding predicted values. The optimal conditions are an ultrasonic frequency of 60 kHz and a variation ratio of ionic liquid (ILs) to water (2:8). Under these conditions, the model predicts the maximum responses for TPC (0.0103 mg GAE/g), DPPH (74.22%), and FRAP (6.09 mg AEAC/g) in *Clinacanthus nutans* leaf extract.

### 3.7 Verification of predictive model

In addition to evaluating significant differences, the model's suitability was assessed through a lack-of-fit analysis. Lack-of-fit analysis is a statistical method utilized to assess the model's suitability and determine how well it fits the obtained data. This analysis also assesses the appropriateness of various models based on response surface methodology [77]. The three dependent variables considered in this study were phenolic content (TPC), free radical scavenging activity (DPPH), and ferric reducing antioxidant power (FRAP), with ultrasonic frequency (kHz) and the variation ratio of the concentration of ionic liquid (ILs) and water (v/v) as the independent variables.

To measure the experimental values with the expected results, a confirmation experiment was conducted using the optimum conditions obtained. As shown in Table 7, the experimental results were very close to the predicted values, indicating a high degree of agreement between the regression model's predicted values and the experimental values. Therefore, response surface methodology can be effectively used to predict the maximum extraction outcomes in terms of phenolic content and antioxidant capacity from *Clinacanthus nutans* leaves. Based on the results from ANOVA, it can be concluded that there are no significant differences between the predicted and experimental values within the 95% confidence interval, as indicated by a p-value of 0.409 ($p > 0.05$).

The paired sample t-test results showed a p-value of 0.440, which is higher than the significance level of 0.05 ($p > 0.05$). This indicates that there is no significant difference between the predicted and experimental values for TPC, DPPH, and

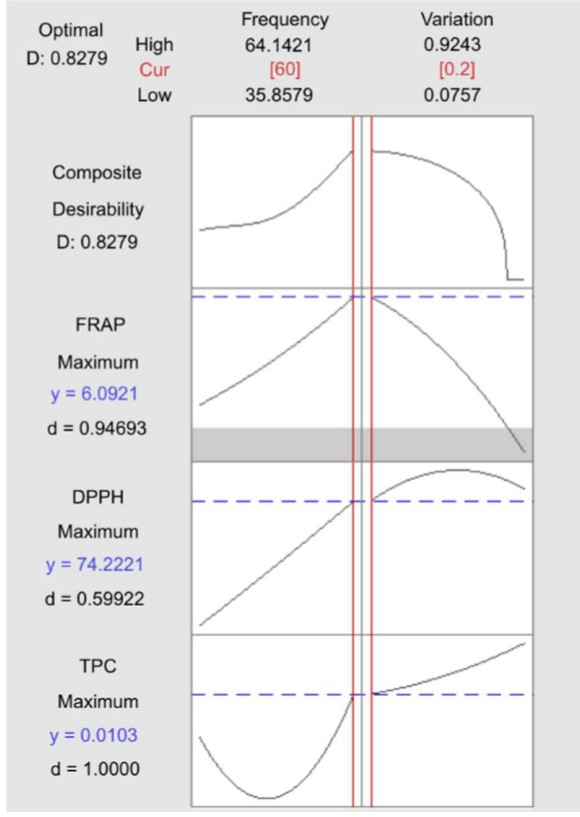

**Fig 10. Optimization desirability plot for ultrasonic-assisted extraction of *Clinacanthus nutans* leaves.** The plot illustrates the predicted maximum responses for TPC, DPPH radical scavenging activity, and FRAP based on the response surface methodology model. Optimal extraction conditions were achieved at an ultrasonic frequency of 60 kHz and an ionic liquid-to-water ratio of 2:8, resulting in TPC (0.0103 mg GAE/g), DPPH (74.22%), and FRAP (6.09 mg AEAC/g). The composite desirability value of 0.8279 indicates a high level of simultaneous optimization for all three responses. The vertical red lines indicate the selected optimal values of the independent variables, while the dashed blue lines represent the predicted maxima for each response variable.

**Table 6. The predicted result of processing optimum parameter from snake grass leaves extracts.**

| Ultrasonic frequency (kHz) | Variation ratio of concentration of ionic liquid and water (v/v) | TPC (mg GAE/g) | DPPH (%) | FRAP (mg AEAC/g) | Desirability |
|---|---|---|---|---|---|
| 60 | 2:8 | 0.0103 | 74.2221 | 6.0922 | 0.8279 |

**Table 7. Comparison between the predicted and experimental values for TPC, DPPH and FRAP under optimum conditions.**

| Dependent variables | Predicted values | Experimental values |
|---|---|---|
| TPC | 0.01 mg GAE/g | 0.01 ± 7.97 x 10$^{-5}$ mg GAE/g |
| DPPH | 74.22% | 95.08 ± 0.57% |
| FRAP | 6.92 mg AEAC/g | 6.31 ± 0.10 mg AEAC/g |

FRAP under optimal conditions. Therefore, the predicted values are validated as consistent and acceptable compared to the experimental values.

### 3.8 Model limitations

The predictive strength of the fitted models was generally acceptable for TPC and FRAP responses. However, a notable limitation was observed in the model for DPPH antioxidant activity. The coefficient of determination ($R^2$) for DPPH was 0.048, indicating a poor fit and suggesting that the model explained only a small portion of the variability in the observed data. This limitation may be attributed to the multifactorial nature of antioxidant activity, which is potentially influenced by variables such as temperature, solvent stability, and compound degradation during sonication factors not included in the current experimental design. Although the quadratic model provided useful trend analysis for DPPH and supported overall process optimization, its predictive capability is limited. For improved modelling accuracy, future studies are encouraged to incorporate additional variables (e.g., extraction time, temperature) or adopt alternative approaches such as non-linear regression or machine learning models (e.g., artificial neural networks), which may better capture complex interactions and enhance model reliability.

### 3.9 Comparative optimal values of TPC, DPPH and FRAP with elsewhere studies

Based on the optimal values of TPC, DPPH, and FRAP in Table 8, a comparison was made between *Clinacanthus nutans* leaf extract and *Strobilanthes crispus* leaf extract, using the same extraction solution (ionic liquid binary solvent). *Strobilanthes crispus* leaves are known for their high phenolic content and antioxidant properties [79]. Therefore, *Strobilanthes crispus* leaf extract was chosen as the control in this study.

The study by Zainol et al. (2020) [80] reports that the phenolic content in *Strobilanthes crispus* leaf extract is comparable to that found by Ghasemzadeh et al. (2014) [78]. Table 9 shows the comparison of parameters used in this study and previous studies. In Zainol et al. (2020), the leaves were extracted using varying concentrations of methanol and acetone for 1 hour in the dark at room temperature before testing antioxidant activity [80] In fact, methanol was successfully efficient to extract total phenolic content in several plants [68]. In contrast, Ghasemzadeh et al. (2014) added methanol to *Strobilanthes crispus* leaves and performed the extraction for 2 hours at room temperature (25°C) [78]. These studies confirm that *Strobilanthes crispus* leaves are suitable as a control for this research.

**Table 8. Comparison between optimal values for TPC, DPPH and FRAP with other extraction process.**

|  | Ultrasonic extraction | Normal soaking extraction | |
| --- | --- | --- | --- |
|  | Current study (optimized parameters) | Ghasemzadeh et al. (2014) [78] | Zainol & Mansor (2019) [7] |
|  | Sabah snake grass | Sabah snake grass | Glass splinter |
| TPC | $0.01 \pm 7.97 \times 10^{-5}$ mg GAE/g | 11.32 mg GAE/g | 9.05 mg GAE/g |
| DPPH | $95.08 \pm 0.57\%$ | 52.3% | 18.88 mg TE/g |
| FRAP | $6.31 \pm 0.10$ mg AEAC/g | N/A | 37.09 mg TE/g |

**Table 9. Comparison of parameters used in this study and previous studies.**

| Studies | Solution used | Method of extraction | Time extraction |
| --- | --- | --- | --- |
| Current study | Deep eutectic solvent (DES) and water | Ultrasonic extraction | 3 hours |
| Ghasemzadeh et al. (2014) [78] | Methanol and hydrochloric acid (HCl) | Reflux (solid-liquid extraction) | Shake for 2 hours and reflux for 2 hours |
| Zainol & Mansor (2019) [7] | Different concentration of methanol and acetone (25%, 50%, 75% and 100%) | Laboratory orbital shaker | 1 hour |

Meanwhile, the optimal parameters in this study (ultrasonic frequency of 60 kHz and a variation ratio of the concentration of ionic liquid and water of 2:8) involved ultrasonic extraction for 3 hours using the binary ionic liquid solvent (DES). However, the phenolic content observed in this study, which used ultrasonic extraction, is significantly lower compared to Ghasemzadeh et al. (2014), which used traditional soaking extraction [78]. While UAE improves the release of bioactive compounds through increased mass transfer due to cavitation effects, such as microjet formation causing surface peeling and erosion [81], there exists a potential trade-off in the stability of sensitive compounds during UAE. The localized hotspots and reactive radicals generated by ultrasonic cavitation can unintentionally degrade heat-sensitive phenolic compounds if the extraction conditions are not carefully controlled. For example, UAE can produce higher extraction yields than maceration, thus the optimization of extraction parameters such as temperature and time is crucial to minimize detrimental effects on compound stability [82]. In contrast, more traditional extraction methods, such as ethanol reflux extraction, tend to operate under milder conditions, which can help preserve the stability of certain antioxidants but may sacrifice extraction efficiency [83]. For instance, Hou et al. (2019) point out that optimized traditional extraction methods may yield higher total phenolic content, although they do not always provide the same antioxidant activity as UAE extracts [84]. In a comparative study, Xu et al. noted that while total phenolic content may be lower with UAE than with conventional methods, the DPPH radical scavenging activity is often higher, indicating that optimizing UAE conditions is key for balancing extraction yield with the preservation of antioxidant activity [85].

Several factors may explain this discrepancy. Ultrasonic extraction relies on high-frequency waves to create cavitation, generating small bubbles in the solution that collapse rapidly, producing strong pressure and sudden temperature changes [15]. The burst of bubbles disrupts plant cell walls, releasing intracellular components and enabling faster extraction [41,86]. However, if this process is not carefully controlled, the released energy can damage sensitive components in the *Clinacanthus nutans* leaf extract. In contrast, traditional soaking extraction does not generate cavitation bubbles; instead, it relies on passive diffusion, where phenolic compounds move from plant cells into the solvent over time [27]. Since no mechanical power or heat is generated from cavitation, the risk of heat degradation and oxidation of phenolic compounds is minimized, leading to higher extraction yields. Additionally, the study by Nurulita et al. (2008) [87] noted that phytochemicals such as alkaloids, tannins, quinones, steroids, flavonoids, triterpenoids, and diterpenes can be extracted through water, as confirmed by Ho et al. (2013) [88]. However, phenolic compounds cannot be effectively extracted with water and require 100% methanol [88]. This could be one reason why the phenolic content in this study was significantly lower compared to the studies by Ghasemzadeh et al. (2014) [78] and Zainol et al. (2020) [80].

According to Table 8, the antioxidant activity (DPPH) in this study (95.075%) was higher than that observed in Ghasemzadeh et al. (2014) (52.3%) [78]. According to Hayouni et al. (2007), the polarity of the solvent influences the extraction of phenolic compounds, with water (higher polarity) attracting more phenolic compounds than less polar solvents (such as methanol) [89]. However, the results from this study contradict this theory, suggesting the need for further research to clarify these findings. Additionally, the study by Zainol and Mansor (2019) showed that the storage period of extracts affects DPPH antioxidant activity [7]. For example, Raya et al. (2015) found that the antioxidant activity of *Clinacanthus nutans* extract decreased from 69.97% on day one to day four of storage [90]. The Ferric Reducing Antioxidant Power (FRAP) values for *Clinacanthus nutans* leaf extract cannot be directly compared with those of other studies due to the use of different standards in each study.

## 4. Conclusion

This study successfully demonstrated the potential of ionic liquid (IL) binary solvents in optimizing the extraction of phenolic compounds and enhancing the antioxidant activity of *Clinacanthus nutans* leaves using ultrasonic-assisted extraction. By employing Peleg's mathematical model, an optimal predicted extraction time of 3 hours was determined, ensuring

maximum yield efficiency while minimizing processing time. However, Response Surface Methodology (RSM) further identified the ideal extraction parameters, revealing that an ultrasonic frequency of 60 kHz and an ionic liquid (IL)-to-water ratio of 2:8 produced the highest total phenolic content (TPC) ($0.01 \pm 7.97 \times 10^{-5}$ mg GAE/g), DPPH antioxidant activity ($95.08 \pm 0.57\%$), and FRAP antioxidant activity ($6.31 \pm 0.10$ mg AEAC/g). These findings confirm that higher ultrasonic frequencies and optimized solvent compositions significantly enhance the release of bioactive compounds, thereby improving extraction efficiency. The utilization of ionic liquid binary solvents presents a promising green chemistry approach for botanical extractions, offering a sustainable alternative to conventional organic solvents. This method can be applied in nutraceutical, pharmaceutical, and cosmetic industries to enhance the bioavailability of plant-derived antioxidants. However, the discrepancy between the predicted TPC maximum values obtained from the Peleg model ($42.56 \pm 0.0003$ mg GAE/g) and RSM optimization is primarily due to the differing objectives and experimental conditions of each approach in which Peleg's model reflects specific experimental data at 50 kHz, while RSM predicts outcomes based on optimized variable combinations. Additionally, the RSM-predicted value may be influenced by model extrapolation or scaling limitations, leading to an unrealistically low estimate. Furthermore, optimizing other processing parameters such as temperature control during ultrasonic extraction could further improve yield efficiency and compound stability. Overall, this study highlights the effectiveness of combining ultrasonic-assisted extraction with ionic liquid binary solvents for maximizing the recovery of phenolic compounds and antioxidants from *Clinacanthus nutans*, paving the way for broader applications in health and wellness industries.

## 5. Future research directions

Future research should focus on high-performance liquid chromatography (HPLC) profiling of *Clinacanthus nutans* extracts obtained under optimized conditions to identify and quantify specific phenolic constituents. To accurately quantify bioactive compounds and establish their correlation with bioactivity, a targeted High-Performance Liquid Chromatography (HPLC) method using UV detection at 280 nm is recommended. This method is recognized as effective for analyzing phenolic compounds given their characteristic UV absorption spectra [91]. Previous research has validated the use of HPLC for the determination of phenolic content, as it allows for the separation and identification of individual components, ultimately enhancing the understanding of their functional efficacy [92]. The implementation of such methodologies will facilitate a greater comprehension of the relationship between extraction parameters and the antioxidant efficacy of plant-derived substances, which is critical for their potential application in nutraceutical and therapeutic fields [93,94]. Future research should also explore various processing parameters to optimize extraction methods by targeting specific active constituents with a longer concoction duration through other kinetic equilibrium model (e.g., combined models that combine different kinetic mechanisms, like a second-order model for initial washing followed by diffusion). Specifically, controlling temperature during sonication can significantly influence the yield and quality of phenolic compounds extracted from plants [92]. Furthermore, the assessment of IL solvents' reusability and recyclability could extend the economic viability and sustainability of extraction processes [95]. Additionally, evaluating the bioavailability of these extracted phenolics *in vivo* is important for substantiating their biological effects and commercial viability [96].

## Supporting information

**S1 File. TPC DPPH FRAP raw data & results.**
(XLSX)

## Acknowledgments

We are grateful to Universiti Kebangsaan Malaysia (UKM) and the Department of Food Sciences, Faculty of Science and Technology, UKM Bangi for allowing this study to be carried out at the Food Pilot Plant.

## Author contributions

**Conceptualization:** Saiful Irwan Zubairi.

**Formal analysis:** Saiful Irwan Zubairi, Tong Yen Suan, Zalifah Mohd Kasim, Nur Huda-Faujan.

**Funding acquisition:** Saiful Irwan Zubairi.

**Investigation:** Saiful Irwan Zubairi, Tong Yen Suan, Zalifah Mohd Kasim.

**Methodology:** Saiful Irwan Zubairi, Nur Huda-Faujan, Ruth Naomi Manuel.

**Project administration:** Saiful Irwan Zubairi.

**Resources:** Tong Yen Suan.

**Software:** Tong Yen Suan.

**Supervision:** Saiful Irwan Zubairi.

**Validation:** Zalifah Mohd Kasim, Nur Huda-Faujan, Ruth Naomi Manuel.

**Visualization:** Zalifah Mohd Kasim.

**Writing – original draft:** Tong Yen Suan.

**Writing – review & editing:** Saiful Irwan Zubairi, Nur Huda-Faujan, Ruth Naomi Manuel.

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
