## [Decision Letter · Decision Letter 0]

Dear Dr. Zubairi,

Thank you for submitting your manuscript to PLOS ONE. After careful consideration, we feel that it has merit but does not fully meet PLOS ONE’s publication criteria as it currently stands. Therefore, we invite you to submit a revised version of the manuscript that addresses the points raised during the review process.

We look forward to receiving your revised manuscript.

Kind regards,

Andrea Mastinu

Academic Editor

PLOS ONE

Journal Requirements:

We express gratitude for the funding provided by Universiti Kebangsaan Malaysia (UKM) under grant ST-2023-043, which enabled us to conduct this study

We express gratitude for the funding provided by Universiti Kebangsaan Malaysia (UKM) under grant ST-2023-043, which enabled us to conduct this study. Furthermore, we extend our appreciation to the Department of Food Sciences, Faculty of Science and Technology, UKM Bangi, for granting us access to their laboratory facilities.

We express gratitude for the funding provided by Universiti Kebangsaan Malaysia (UKM) under grant ST-2023-043, which enabled us to conduct this study

4. We note that your Data Availability Statement is currently as follows: All relevant data are within the manuscript and its Supporting Information files

Reviewers' comments:

Reviewer's Responses to Questions

**Comments to the Author**

1. Is the manuscript technically sound, and do the data support the conclusions?

Reviewer #1: Partly

Reviewer #2: Partly

2. Has the statistical analysis been performed appropriately and rigorously?

Reviewer #1: Yes

Reviewer #2: Yes

3. Have the authors made all data underlying the findings in their manuscript fully available?

Reviewer #1: Yes

Reviewer #2: No

4. Is the manuscript presented in an intelligible fashion and written in standard English?

Reviewer #1: Yes

Reviewer #2: No

Reviewer #1: The present study describes the optimization of an extraction method for the recovery phenolic compounds from Clinacanthus nutans plant using DES as extractor medium. Although the authors present several pharmacological properties in introduction, they did not correlate them with specific phenolic compounds. Thus, they performed simple photometric assays as TPC, DPPH and FRAP to carry out the optimization. Are there any correlation of these measurements with specific biological or pharmacological activity? Although the the extraction parameters are optimized, the usefulness of extraction is uncertain.

Specific comments:

1. Please explain why did you use this specific DES? A wide range of DES solvents are available, Did you perform prelimenary experiments?

2. Which is the usefulness of Peleg’s model?

3. Why did you use an other plant material as benchmark?

Reviewer #2: This manuscript explores the optimization of ultrasonic-assisted extraction (UAE) of phenolic compounds from Clinacanthus nutans leaves using ionic liquid (IL) binary solvents. It applies Peleg’s kinetic model to determine extraction time and response surface methodology (RSM) to optimize extraction parameters. The study addresses the need for green and efficient extraction methods and finds optimal conditions yielding high phenolic content and antioxidant activity. While the research is timely and relevant, several areas could be improved in terms of scientific justification, clarity, and methodological rigor before can be considered for publication. The suggestions for improvement are listed below:

Abstract

• Line 29: The reported TPC value of "0.01 ± 7.97 × 10⁻⁵ mg GAE/g" appears inconsistent with the earlier reported value of "42.556 ± 0.0003 mg GAE/g" (Line 25). This contradiction needs clarification to avoid reader confusion.

Introduction

• Line 45: The section provides a comprehensive ethnopharmacological background but lacks a clear identification of the research gap.

• Line 91: Consider explicitly stating the novelty of applying IL binary solvents with Peleg’s model and RSM in combination, as this is the unique contribution of the study. Also, justification why Peleg’s model was chosen for deciding the optimum extraction time instead of combining the independent variable of time in the CCRD experimental design.

• Please clearly justify the needed of using Peleg’s model over other kinetic models. How does it offer superior insight for this type of extraction?

Materials and Methods

• Line 94: The addition of Strobilanthes crispus as a benchmark is interesting but underexplained. The rationale for its inclusion should be made explicit.

• Line 117: The drying process may affect compound stability. Please clarify how compound degradation due to heat was prevented or controlled.

• Line 180: No information is provided about the temperature control during UAE. This is critical because ultrasound generates heat which can degrade phenolic compounds. This should be addressed.

• Line 254: While ANOVA is used, consider reporting effect sizes or regression coefficients to provide deeper insights into variable impact.

Results and Discussion

• Line 306: The R² values are low (especially for DPPH: 0.048), which significantly undermines model predictability. The manuscript should discuss limitations of the model and perhaps consider model refinement or alternate modeling approaches.

• Line 408: There is conflicting interpretation regarding whether lower or higher ultrasonic frequencies enhance FRAP. Authors should clarify or reanalyze the statements to maintain consistency.

• Line 492: The comparison between UAE and traditional extraction methods is important, but should discuss the potential trade-off between yield and compound preservation due to ultrasound-induced degradation.

• While the use of ILs and Peleg’s model is appropriate, deeper discussion of solvent interaction mechanisms (e.g., hydrogen bonding, viscosity effects) should be integrated to enrich the explanation beyond statistical outcomes.

Conclusion

• Line 537: The suggestion for future research is valid but broad. It could be strengthened by proposing a more specific HPLC analysis plan or mentioning which phenolic compounds are suspected to be dominant.

• Line 541: Temperature control during ultrasonic extraction is listed as a future consideration—this should also be discussed in the Methods or Results section.

Overall

• Keywords (Line 42): Consider revising keywords to be more distinct from the title to enhance discoverability (e.g., “green solvent extraction,” “Clinacanthus nutans bioactives”).

• Nomenclature: The scientific name Clinacanthus nutans should be italicized throughout the text.

• Figures and Tables: Ensure all figures are referenced in the text in sequential order. Currently, some figure explanations (e.g., Figure 10) appear without context.

Refrences

• Alphabetize the reference list and ensure uniformity in journal naming (italicization, abbreviations).

• Some citations are quite dated (e.g., 2000–2009). Consider adding more recent references (within the last 5 years) to support recent claims about DES and ultrasonic-assisted extraction mechanisms

**Do you want your identity to be public for this peer review?** For information about this choice, including consent withdrawal, please see our Privacy Policy

Reviewer #1: No

Reviewer #2: No

---

## [Author Response · Author response to Decision Letter 1]

21 Apr 2025

Lists of correction files are attached in the revised submission files. Thank you

---

## [Decision Letter · Decision Letter 1]

Optimization of Ultrasonic-Assisted Extraction of Phenolic Compounds from Clinacanthus nutans using Ionic Liquid (ILs) Binary Solvent: Application of Peleg’s Model and Response Surface Methodology

PONE-D-25-12599R1

Dear Dr. Zubairi,

We’re pleased to inform you that your manuscript has been judged scientifically suitable for publication and will be formally accepted for publication once it meets all outstanding technical requirements.

Kind regards,

Andrea Mastinu

Academic Editor

PLOS ONE

Additional Editor Comments (optional):

Reviewers' comments:

Reviewer's Responses to Questions

**Comments to the Author**

Reviewer #2: All comments have been addressed

2. Is the manuscript technically sound, and do the data support the conclusions?

Reviewer #2: Yes

3. Has the statistical analysis been performed appropriately and rigorously?

Reviewer #2: Yes

4. Have the authors made all data underlying the findings in their manuscript fully available?

Reviewer #2: Yes

5. Is the manuscript presented in an intelligible fashion and written in standard English?

Reviewer #2: Yes

Reviewer #2: I find no remaining substantive concerns, and I am satisfied with the scientific rigor and clarity of the revised version.

**Do you want your identity to be public for this peer review?** For information about this choice, including consent withdrawal, please see our Privacy Policy

Reviewer #2: No

---

## [Editor Report · Acceptance letter]

PONE-D-25-12599R1

PLOS ONE

Dear Dr. Zubairi,

I'm pleased to inform you that your manuscript has been deemed suitable for publication in PLOS ONE. Congratulations! Your manuscript is now being handed over to our production team.

Kind regards,

on behalf of

Dr. Andrea Mastinu

Academic Editor

PLOS ONE